# Research on Water-Level Recognition Method Based on Image Processing and Convolutional Neural Networks

**Gang Dou** [1,2], **Rensheng Chen** [1,3,*], **Chuntan Han** [1,2], **Zhangwen Liu** [1] and **Junfeng Liu** [1]

1    Qilian Alpine Ecology and Hydrology Research Station, Northwest Institute of Eco-Environment and Resources, Chinese Academy of Sciences, Lanzhou 730000, China; dougang20@mails.ucas.ac.cn (G.D.); hancht@lzb.ac.cn (C.H.); zwliu@lzb.ac.cn (Z.L.); liujfzyou@lzb.ac.cn (J.L.)
2    University of Chinese Academy of Sciences, Beijing 100049, China
3    College of Urban and Environment Sciences, Northwest University, Xi'an 710127, China
*    Correspondence: crs2008@lzb.ac.cn; Tel.: +86-0931-4967166

**Abstract:** Water level dynamics in catchment-scale rivers is an important factor for surface water studies. Manual measurement is highly accurate but inefficient. Using automatic water level sensors has disadvantages such as high cost and difficult maintenance. In this study, a water level recognition method based on digital image processing technology and CNN is proposed. For achieving batch segmentation of source images, the coordinates of the water ruler region in the source image and characters' region and the scale lines' region on the ruler are obtained by using image processing algorithms such as grayscale processing, edge detection, and the tilt correction method based on Hough-transform and morphological operations. The CNN is then used to identify the value of digital characters. Finally, the water level value is calculated according to the mathematical relationship between the number of scale lines detected by pixel traversal in the binarized image and the value of digital characters. This method is used to identify the water levels of the water ruler images collected in the Hulu watershed of the Qilian Mountains in Northwest China. The results show that the accuracy compared with the actual measured water level reached 94.6% and improved nearly 24% compared to the template matching algorithm. With high accuracy, low cost, and easy deployment and maintenance, this method can be applied to water level monitoring in mountainous rivers, providing an effective tool for watershed hydrology research and water resources management.

**Keywords:** image recognition; CNN; water level identification; water ruler; Hough transform

## 1. Introduction

There are many countries with massive reservoirs, widely distributed rivers and frequent flood disasters [1,2]. The establishment of a river reservoir water level monitoring system has long been an important means to manage river basins and reduce the risk of flooding disasters and ensure the safety of waterways [3]. At the same time, the water level is the key data to be collected in water resources monitoring. A common method of water level observation is manually checking the visual readings. This method requires real-time on-site observation, which is labor-intensive, inefficient and even dangerous. Automatic observation uses sensors to collect analog quantities that characterize the water level, and then convert them into water level data [4,5]. Pressure sensors, rangefinders, ultrasonic, radar and optical sensors are commonly used. The disadvantages are also obvious, such as the high cost of setup, the susceptibility to changes in water quality, the need for frequent manual adjustment of equipment, the susceptibility to external environmental interference, and so on [6]. This has led many monitoring units to opt for low-cost cameras, the use of which can provide images related to water levels, and it is possible to develop it into a computer vision system for remote monitoring and visual inspection of river sites [7].

In the past 30 years, digital image processing technology has evolved rapidly and has been widely used [8]. Computers have gradually replaced the human brain's cognition of

visualizing scenes through image processing [9]. The method based on image processing has the advantages of high efficiency and degree of automation. Krizhevsky et al. [10] have built AlexNet based on deep learning theory, and won the championship in the image classification competition ILSVRC. The AlexNet showed absolute advantages in both classification effect and computing speed. The rise of CNN based on deep-learning algorithms provides new methods and driving forces for image processing [11]. After that, ZFNet [12], VGG-Net [13], GoogleNet [14] and ResNet [15], which won the ILSVRC competition, are all image classification algorithms based on deep learning. The classification accuracy and computational speed of these algorithms are constantly improving, which drives the development of computer vision fields. Artificial intelligence algorithms such as clustering, SVM, KNN, decision tree, random forest and other machine learning algorithms, as well as deep learning algorithms with a neural network as the core, have shown great potential for application in the field of geography and hydrology and water resources in recent years. Yu et al. [16] proposed a new method of water extraction based on CNN and logistic regression classifier as a new method for water body extraction, and using ANN and SVM methods for comparison, the final results show that the deep learning method has the highest accuracy. Bai et al. [17] established a multi-scale deep feature learning method for predicting the incoming flow in the Three Gorges reservoir area and confirmed the feasibility of the deep learning method for hydrological forecasting. Sabbatini et al. [18] propose an automated CV solution capable of detecting and calculating river water levels with a frame captured by a V-IoT device as input, and a high degree of automation is achieved in the image acquisition and pre-processing stages, but the water level calculation stage is still not intelligent enough. Jafari et al. [19] used a deep learning-based semantic segmentation technique to identify reference objects in videos and images to estimate the water level over time; RamKumar et al. [20] used a feature matching algorithm to find the feature points corresponding between the captured image and the reference image, and then estimated and plotted the flood lines. The accuracy of the two methods above is susceptible to the influence of the surrounding environment.

In order to solve the problems existing in the current water level monitoring methods and make the process of monitoring more intelligent, a new water level image recognition method is proposed in this study. First, the source image is pre-processed by using digital image processing techniques such as grayscale transformation, edge detection, tilt correction based on Hough transform and a morphological algorithm. The coordinates of the areas in the water ruler and the digital characters and the scale lines are obtained, then the source image is segmented in a batch according to these coordinates. The CNN is then designed to identify the values of digital characters. Finally, the water level value is calculated according to the mathematical relationship between the number of scale lines detected by pixel traversal in the binarized image and the value of characters.

This paper starts with an introduction that compares the previous work done and the research progress in the field, The materials and methods section show the key techniques and methods used in this study. In the results section, with the same sample of photos, using the manual observation as the standard value, the recognition results of the intelligent algorithm proposed in this study and the template matching algorithm are compared. Finally, a discussion and conclusion of the whole study is given.

## 2. Materials and Methods

### 2.1. Study Region and Data Acquisition

The water ruler source images in this study were taken in a hydrographic cross-section of the Hulu watershed (38.2 °N, 99 °E), which is located in the upper reaches of the Hei River in the central part of the Qilian Mountains [21]. It comprises an area of ~23.1 km$^2$ and the elevation range between 2960 and 4820 m. The Hulu watershed belongs to a continental climate, the average annual temperature is $-0.3$ °C and the average annual precipitation is 599.8 mm [22]. The average annual runoff was 10,035,203.71 m$^3$ from 2010

to 2015. Precipitation in this region is frequent and 89% of the precipitation falls during the wet season from May to September [23,24].

This hydrographic section was built in June 2010 and has an elevation of about 2960 m, which was represented by "Stream-gauging stations" in Figure 1b. The section is generally trapezoidal in shape, but is divided into two parts, which guarantees the accuracy of measurement at low water level, but also guarantees that flood water can pass smoothly. Figure 2 shows the detailed parameters of this hydrographic cross-section. The total width of the bridge deck is 3.0 m, the width of the bottom section is 1.15 m, and the width of the middle section is 2.6 m. Moreover, the total length of the middle section is 9.4 m, and the total external length of the section is 10.1 m.

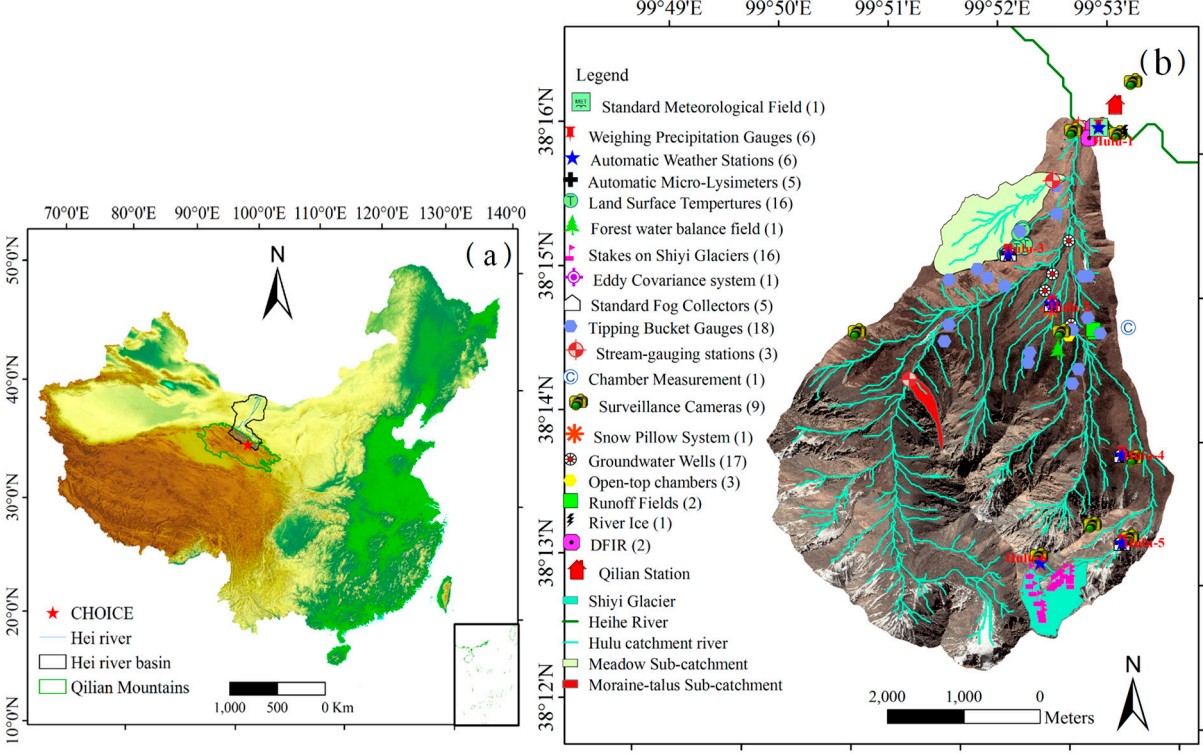

**Figure 1.** (**a**) The geographical location of Hulu watershed. (**b**) The distribution of instruments and observatories of the Hulu watershed [22].

Earlier in the process, used an automatic water level logger were used to measure water level of the cross-section shown in Figure 2, which has some drawbacks, such as high equipment cost, high environmental cost, and the equipment is easily affected by the environment thereby does not function properly. Specifically, the current widely used water level logger (HOBO U20-001-01, Onset) needs to be placed with one inside and another outside the water well, which is marked with blue text in Figure 2b, and the unit price of this water level meter is about 5000 CNY or more. In contrast, the use of an automatic continuous filming infrared camera plus a ruler only costs about 3000 CNY. In addition to the cost of the common hydrographic section of about 130,000 CNY, the method of water level monitoring using a water level meter requires the construction of an additional well, which costs about 20,000 to 30,000 CNY to build. This is a comparison of the environmental cost. In what is a serious situation, the wells often silt up and need to be cleaned manually (Figure 3), which leads to more labor and economic costs.

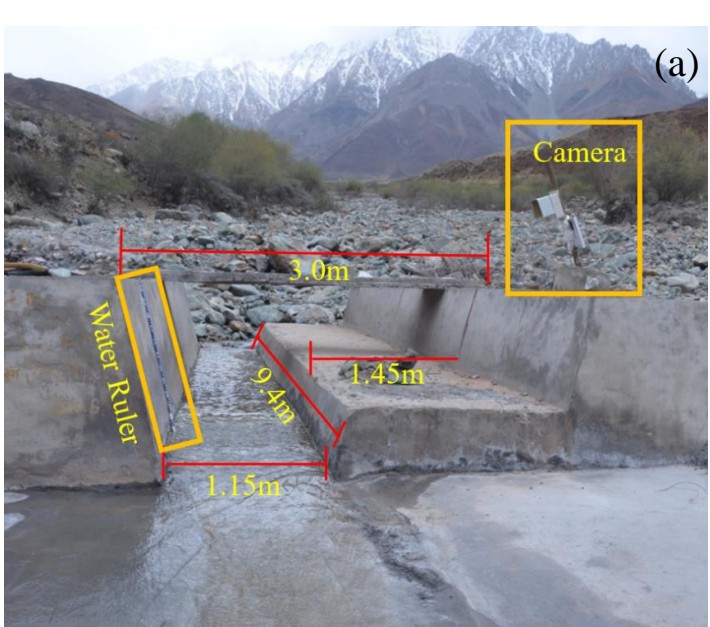
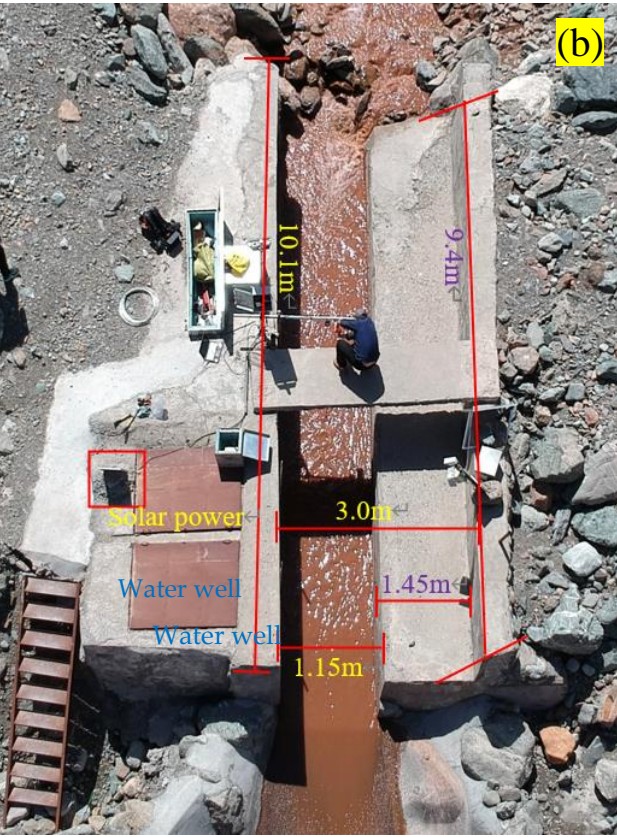

**Figure 2.** The hydrographic cross-section located in the Hulu watershed of the Qilian Mountains. (**a**) Cross-sectional view. (**b**) Vertical view. The camera in this figure is an infrared sensor camera called LTL5120A, which can automatically shoot continuously according to the set time interval, and the resolution of the photos taken is 2560 × 1920.

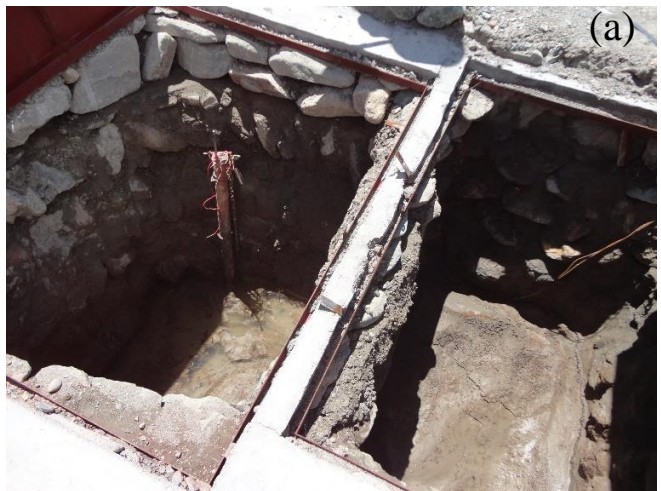
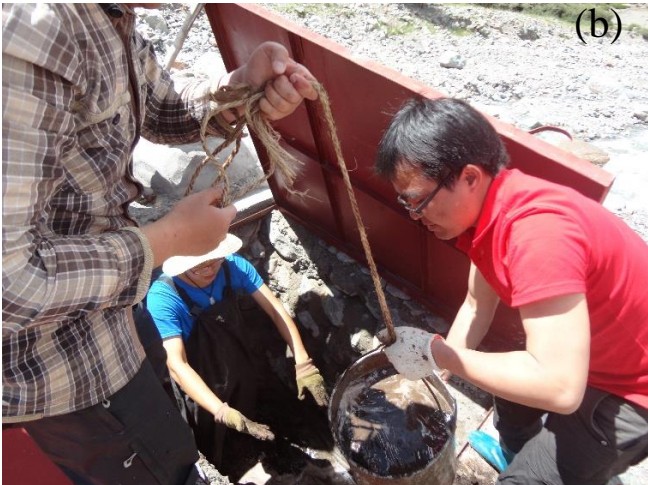

**Figure 3.** Two wells constructed of the hydrographic cross-section. (**a**) The silted-up wells. (**b**) Manual dredging from the water wells.

Figure 4 illustrates the schematic diagram of the collection system for the water level image. A water ruler was set on the cross-section of the river, where the video camera shot automatically. The camera is powered by a solar powered battery. The memory card in the interior of the camera is used to collect photos. These image data are then passed the DTU to the server or imported directly into the server by SD card.

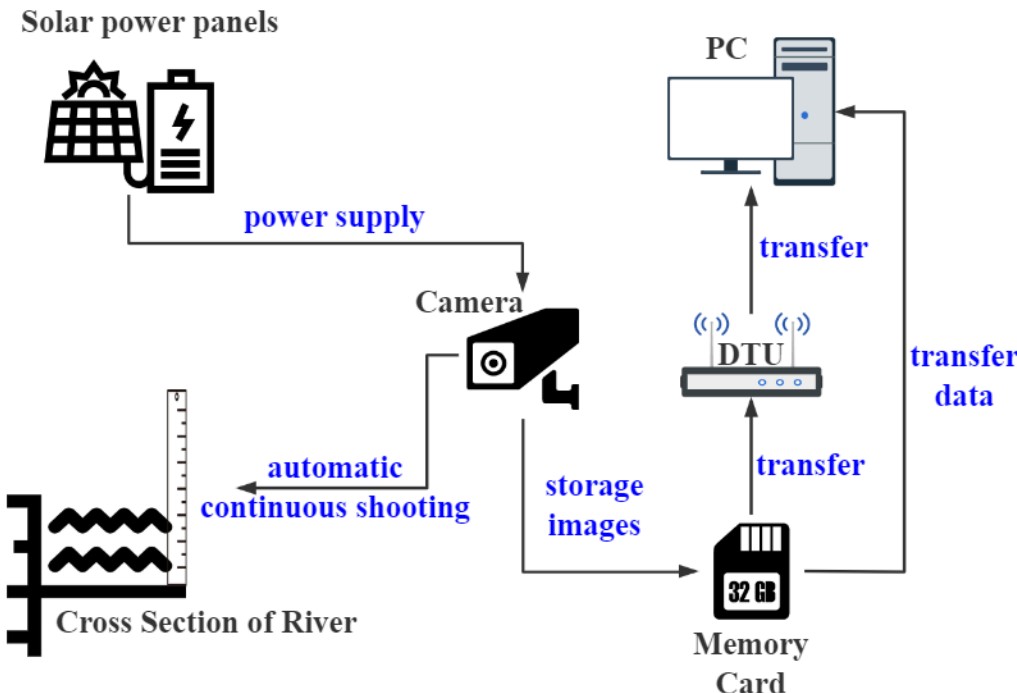

**Figure 4.** System schematic diagram for the water level image in the Hulu watershed.

*2.2. Processing Flow of the Water Level Image Recognition System*

The water level image recognition system was based on image processing and CNN. Figure 5 shows the schematic diagram of the water level image recognition system, which can be divided into three stages: image preprocessing, intelligent identification and water level calculation.

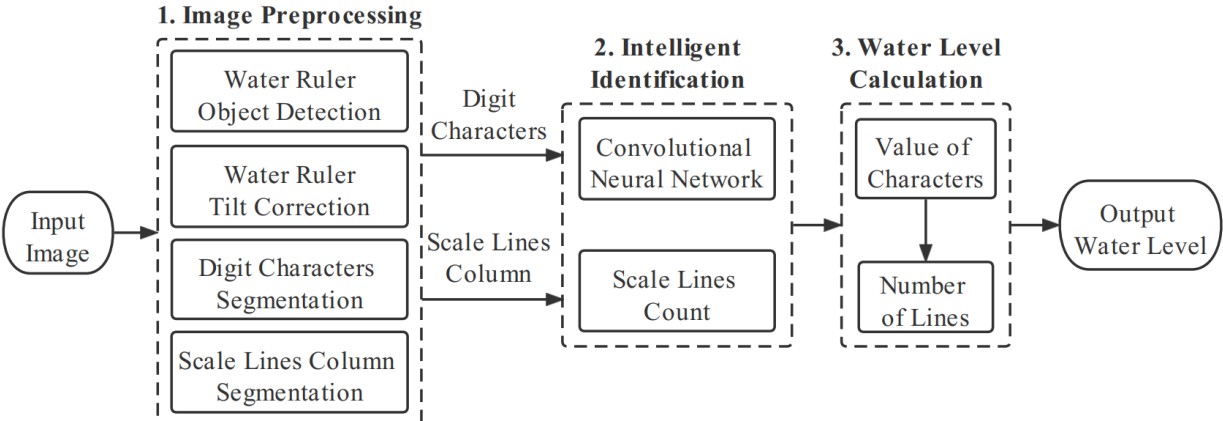

**Figure 5.** Overview of water level image recognition system.

Diverse methods are used in different stages of this intelligent recognition algorithm. In the first stage of image preprocessing, for the same group of photos, since the camera position is fixed, once the coordinates of the water scale area and the character area as well as the scale lines area have been determined in a single photo, all photos in the same group can be split in batches based on these coordinates. The algorithms for digital image processing include filtering, noise reduction, grayscale transformation, binarization, edge detection and Hough transform and projection are used for obtaining the digital characters and the water ruler containing only the area of scale lines mark. In the second stage of intelligent recognition, in order to obtain the values of digital characters and the count of scale lines, a CNN is designed to identify these characters and a method of pixel traversal

is used to identify the scale lines. Finally, in the last stage, the water level can be calculated by the mathematical relationship between the value of characters and the number of scale lines.

### 2.3. Gray Level Transformation

### 2.3.1. Graying

In the RGB model, the color of the pixel at the spatial position $f(x,y)$ uses the $R$ component $R(x,y)$, $G$ component $(x,y)$ and $B$ component $B(x,y)$ of the pixel. All of the three are expressed together [25]. The range of each component is $[0, 255]$, so a pixel can have a color change range of 16.58 million ($255 \times 255 \times 255$). However, each pixel of the grayscale image can only be represented by one grayscale value, that is, a grayscale image can represent most features of the image with fewer data. The process of transforming a color image into the grayscale image is called image grayscale processing, and it can greatly improve the execution efficiency of subsequent algorithms [26]. In this study, the weighted average algorithm of $R$, $G$ and $B$ components of the color image are used to realize grayscale processing. The algorithm proves that the grayscale image obtained by the weighted average grayscale method is better [27]. The formula of the weighted average grayscale method is:

$$f(x, y) = 0.2989R(x, y) + 0.5870G(x, y) + 0.1140B(x, y) \tag{1}$$

Figure 6 shows the variation of the source image to grayscale image.

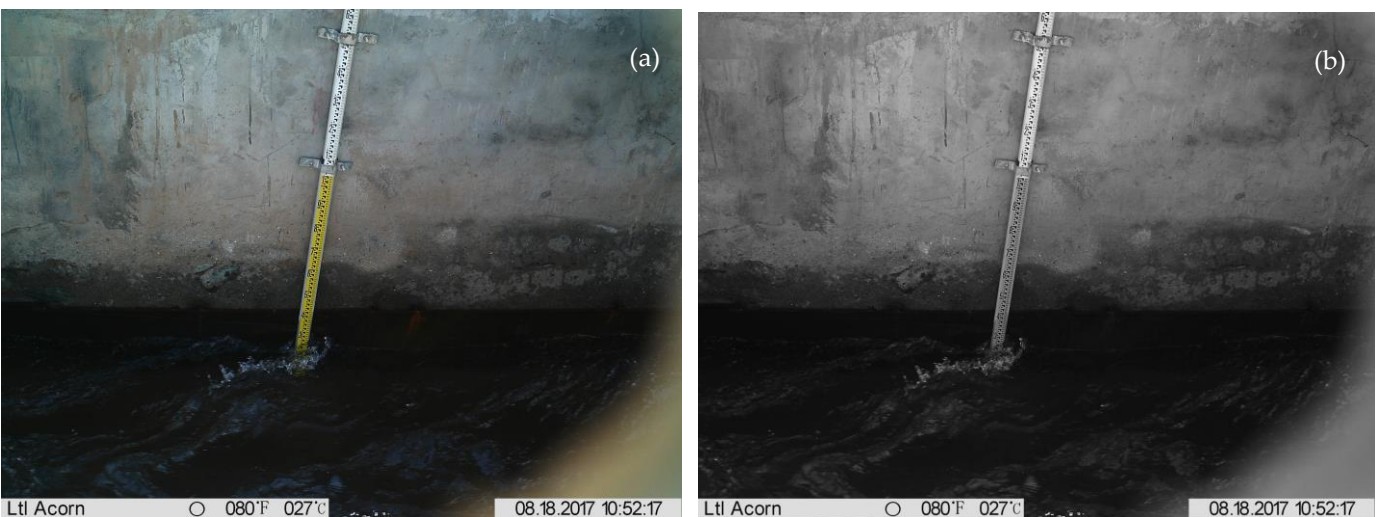

**Figure 6.** (**a**) Original RGB image from the video camera. (**b**) Graying image from (**a**).

### 2.3.2. Binarization

The image binarization operation is used to divide pixels into targets and background [28]. The image shows a distinct black and white effect after binarization, which is currently the most widely used technology in image segmentation. The idea of image binarization algorithm is as follows: assuming that the gray level range in a gray image is (0, 255), then the gray value of each pixel in the image is $f(x, y)$, $f(x, y) \in \{0,1 \dots 255\}$. Assuming that the threshold is $T(0 \leq T \leq 255)$, then:

$$g(x,y) = \begin{cases} 0, & f(x,y) \leq T \\ 255, & f(x,y) > T \end{cases} \tag{2}$$

$g(x,y)$ represents the value of each pixel of the image after binarization.

If $g(x,y) = 255$, it means the point is the target, otherwise the point is the background. The key of binarization is the selection of the threshold value. This research uses the

OTSU [29] method (the Maximum Between-Class Variance Method) in the global threshold method. The idea of this algorithm is similar to clustering [30].

Algorithmic steps:

1. Traverse all the pixels of the image and count the histogram of the gray distribution.
2. Normalize the histogram and set the ratio of the number of pixels with gray value *i* to the total number of pixels as $p(i)$.
3. Assuming that the current threshold is *t*, the normalized histogram can calculate the target pixel ratio $\omega_0$. the normalized histogram can calculate the target pixel ratio $\omega_1$, under the current division, as well as the average gray level of the target area $\mu_0$. under the current division, as well as the average gray level of the target area $\mu_1$.

$$\omega_0(t) = Pr(C_0) = \sum_{i=t+1}^{255} p(i) \tag{3}$$

$$\mu_0(t) = \sum_{i=t+1}^{255} ip(i)/\omega_0(t) \tag{4}$$

$$\omega_1(t) = Pr(C_1) = \sum_{i=0}^{t} p(i) \tag{5}$$

$$\mu_1(t) = \sum_{i=0}^{t} ip(i)/\omega_1(t) \tag{6}$$

The corresponding class variance is:

$$\sigma_0^2 = \sum_{i=1}^{k}(i - \mu_0)^2 Pr(i \mid C_0) = \sum_{i=1}^{k}(i - \mu_0)^2 p_i \ / \ \omega_0 \tag{7}$$

4. To make the intra-class variance the smallest and the inter-class variance the largest, it is equivalent to making $g(t) = \omega_0(t)\omega_1(t)(\mu_0(t) - \mu_1(t))^2$ the largest. OTSU, introduced in the paper, uses the largest between-class variance:

$$\sigma_B^2 = \omega_0(\mu_0 - \mu_T)^2 + \omega_1(\mu_1 - \mu_2)^2 = \omega_0\omega_1(\mu_1 - \mu_0)^2 \tag{8}$$

5. Traverse all the values of T from 0 to 255 to find the value of t that maximizes $g(t)$, that is, the global threshold of the image.

Figure 7 shows the effect of edge detection and binarization in turn.

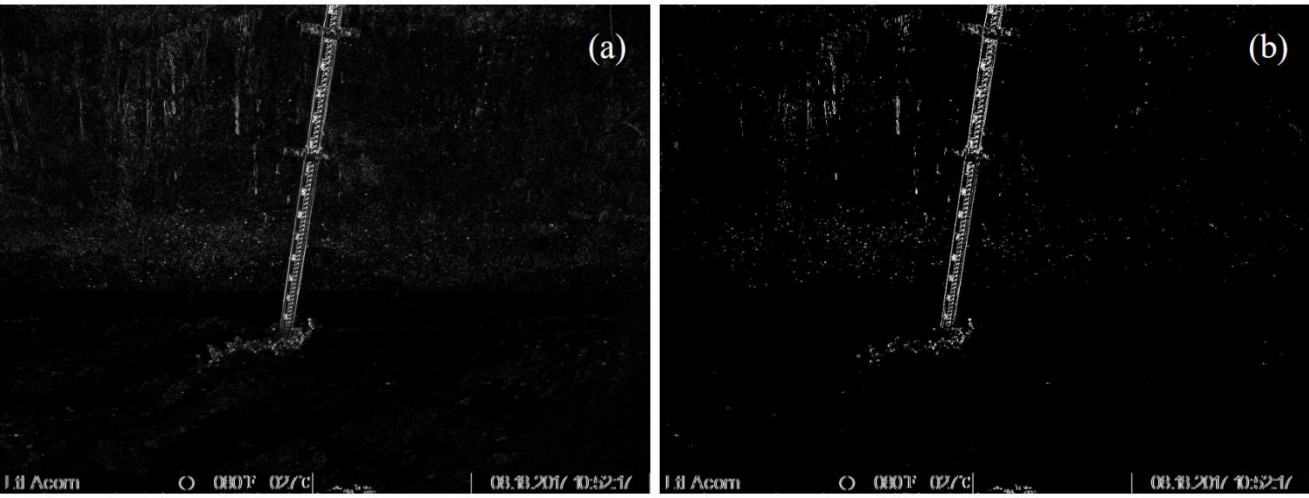

**Figure 7.** (**a**) Sobel-Operated image from Figure 6b. (**b**) The OTSU method for binarization operation from (**a**).

*2.4. Morphological Processing*

The binarized image may also have some noise that interferes with the judgment. It is necessary to further filter this noise through morphological operations. This process is

called denoising. Morphological image processing is a set of non-linear operations related to the shape or morphology of features in an image [31]. In morphological operations, a structural element (a small template image) is used to probe the input image. The working principle of the algorithm is to locate the structural elements in all possible positions in the input image and compare them logically with the corresponding neighborhood of the pixel by the set operator. Commonly used morphological operations or filters include the dilation and erosion of binary images, opening and closing operations, skeletonization, morphological edge detectors, etc.

### 2.4.1. Dilation and Erosion

Dilation and erosion are the basic operations of morphology, and they are also two dual operations [32]. The main purpose is to eliminate the interference of noise. Dilation is the process of incorporating the background points interlinked to the object with the object, and enlarging the boundary to the outside, which can be used to fill the void in the object. Erosion, corresponding to dilation, is a process of eliminating boundary points and constringing the boundary to the inside, which can be used to eliminate the small points and meaningless objects.

The mathematical formula is:

$$X = E \odot B = \{x : B(x) \in E\} \tag{9}$$

$$Y = E \oplus B = \{y : B(y) \cap E \neq \Phi\} \tag{10}$$

### 2.4.2. Dilation and Erosion

Opening and closing operations are related to expansion and erosion operations [33]. They are composed of the combination of expansion and erosion operations and set operations (union, intersection, complement, etc.). They are both dual operations.

Opening operation: The image is eroded first and then dilated.

$$A \bigcirc S = (A \Theta S) \oplus S \tag{11}$$

Effect: It is used to eliminate small objects, smooth the boundary of the shape, and does not change its area. It can remove small particle noise and break the adhesion between objects.

Closing operation: The image is dilated first and then eroded.

$$A \bullet S = (A \oplus S) \Theta S \tag{12}$$

Effect: It is used to fill small holes in objects, connect neighboring objects, and connect broken contour lines.

The effect of the opening and closing operation is shown in Figure 8.

It can be seen that the opening and closing operation further eliminates some irrelevant noise. Horizontal expansion and vertical expansion processing is then performed sequentially on the image, as shown in Figure 9:

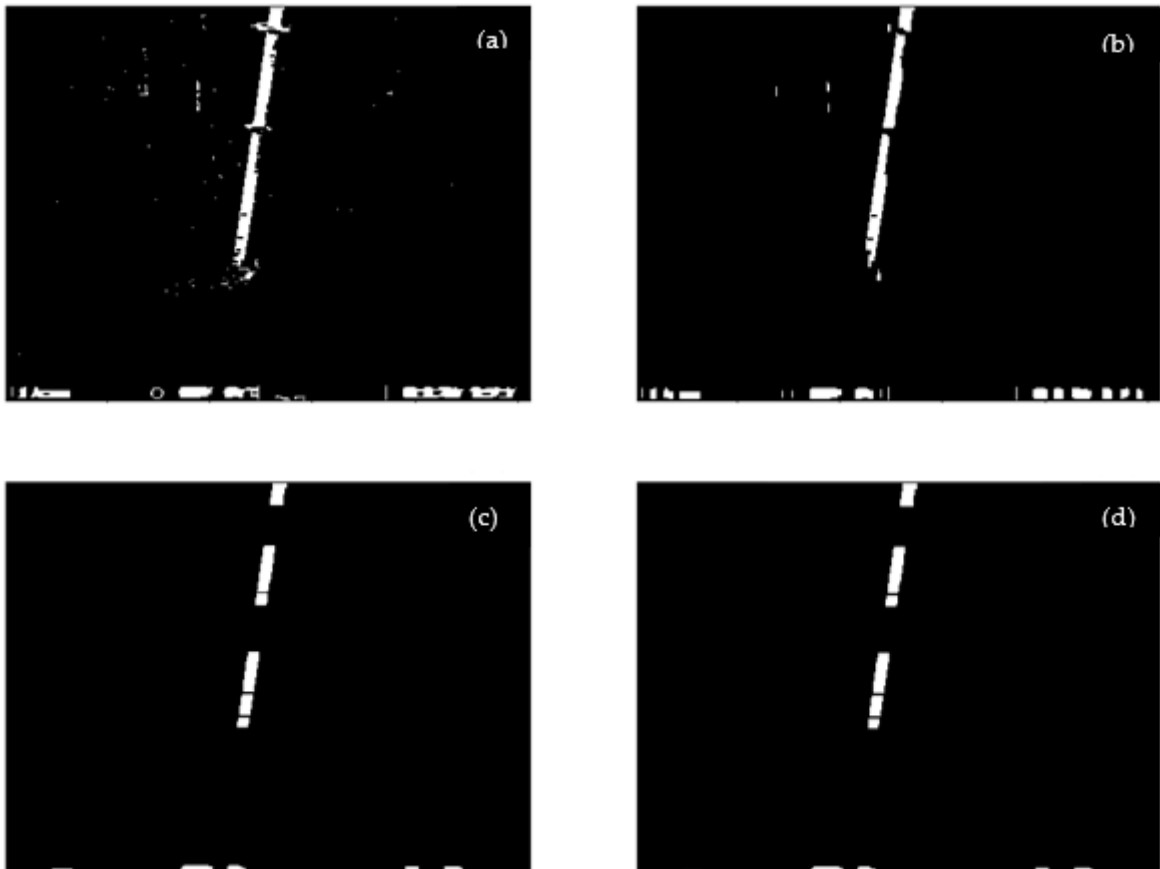

**Figure 8.** The effect after the opening and closing operation. (**a**) The image from Figure 7b by closing operation. (**b**) The image from (**a**) by horizontal opening operation. (**c**) The image from (**b**) by vertical opening operation. (**d**) The image from (**c**) by horizontal opening operation.

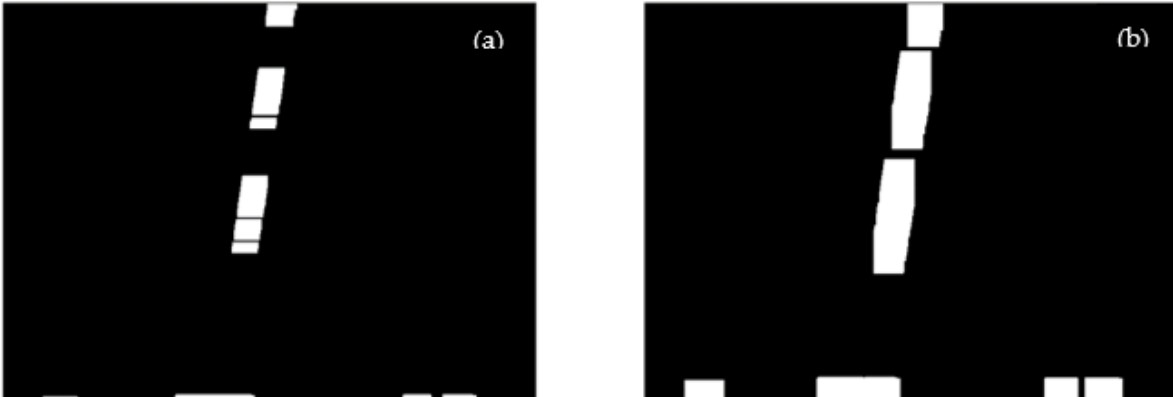

**Figure 9.** The effect after expansion operation. (**a**) The image in Figure 8d by horizontal expansion. (**b**) The image from (**a**) in this figure by vertical expansion.

*2.5. Extraction of Regions of Interest*

2.5.1. Edge Detection

The edge detection algorithm based on the Canny operator was proposed by John Canny [34] in 1986 and is still one of the most classical and advanced algorithms in image edge detection. Compared with operators such as Sobel and Prewitt, the Canny operator has made further refinement and more accurate positioning in terms of surface effects. Operators such as Sobel and Prewitt have the following shortcomings: the gradient

direction of the edge is not fully utilized and the resulting binary image is simply processed with a single threshold. The Canny algorithm made improvements based on these two points and proposed non-maximum suppression based on the edge gradient direction and the Double-threshold hysteresis threshold processing.

General standards for edge detection include detecting as many edges as accurately as possible, these detected edges should be precisely positioned and centered, that an edge should be marked only once, and that the noise should not produce false edges.

Canny uses the variational method that has become one of the most significant edge detection algorithms due to its advantages in meeting the three standards of edge detection and the simple implementation process. The steps of the edge detection algorithm [35] are as follows:

1. We use a Gaussian filter to convolve the image in order to filter out noise and smooth the image to prevent the false detection caused by noise. The convolution kernel scale of $3 \times 3$ or $5 \times 5$ is commonly used. The following formula is the generating equation of the Gaussian filter kernel with a size of $(2k + 1) \times (2k + 1)$:

$$H_{ij} = \frac{1}{2\pi\sigma^2} exp\left( \frac{(i - (k+1))^2 + (j - (k+1))^2}{2\sigma^2} \right); 1 \leq i, j \leq (2k+1) \tag{13}$$

If a $3 \times 3$ window in the image is A and the pixel to be filtered is $e$, after Gaussian filtering, the brightness value of pixel $e$ is:

$$e = \mathrm{H} * \mathrm{A} = \begin{bmatrix} h_{11} & h_{12} & h_{13} \\ h_{21} & h_{22} & h_{23} \\ h_{31} & h_{32} & h_{33} \end{bmatrix} * \begin{bmatrix} a & b & c \\ d & e & f \\ g & h & i \end{bmatrix}$$

$$= sum\left( \begin{bmatrix} a \times h_{11} & b \times h_{12} & c \times h_{13} \\ d \times h_{21} & e \times h_{22} & f \times h_{23} \\ g \times h_{31} & h \times h_{32} & i \times h_{33} \end{bmatrix} \right) \tag{14}$$

where $*$ is the convolution symbol, and *sum* means the sum of all elements in the matrix.

2. The magnitude and direction of the ladder are calculated to estimate the edge strength and direction at each point.

$$G(x, y) = \sqrt{G_x^2(x, y) + G_y^2(x, y)} = |G_x| + |G_y| \tag{15}$$

$$\theta = arc \tan\left(G_y / G_x\right). \tag{16}$$

where $G$ is the gradient strength, the $\theta$ is the gradient direction, $Gx$ is the first derivative value in the horizontal direction, $Gy$ is the first derivative value in the vertical direction.

According to the formula, the gradient and direction of pixel e can be calculated.

Figure 10 is the distribution of the gradient vectors, azimuth angle and edge directions on the centroids.

3. Non-maximum Suppression Non-Maximum Suppression is an edge thinning technique which can help suppress all gradient values other than the local maximum to 0. According to the gradient direction, the gradient amplitude is suppressed by Non-Maximum Suppression to eliminate the stray response caused by edge detection. In essence, this operation is a further refinement of the results of the Sobel and Prewitt operators for meeting the third standard. The algorithm of non-maximum suppression for each pixel in the gradient image is: (1) Compare the gradient intensity of the current pixel with two pixels along the positive and negative gradient direction (not the edge direction). (2) If the gradient intensity of the current pixel is the largest

compared with the other two pixels, the pixel remains as an edge point, otherwise, the pixel will be suppressed. Generally, for more accurate calculation, linear interpolation is used between two adjacent pixels across the gradient direction to obtain the pixel gradient to be compared.

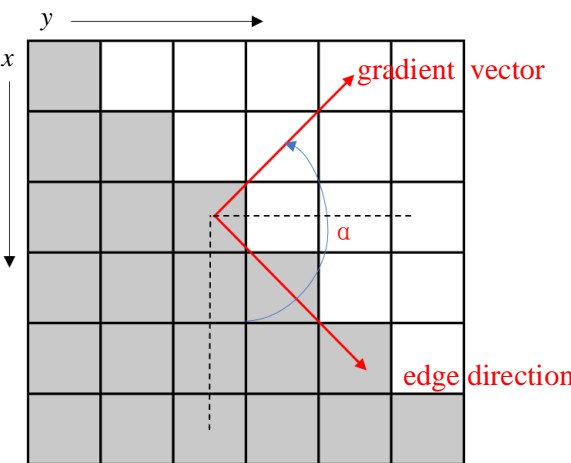

**Figure 10.** The gradient vector, azimuth angle and edge direction of the center point. The edge of any point is orthogonal to the gradient vector.

As is shown in the Figure 11, the gradient is divided into eight directions, namely E, NE, N, NW, W, SW, S, SE. Among them, 0 represents $0° \sim 45°$, 1 represents $45° \sim 90°$, 2 represents $-90° \sim -45°$, and 3 represents $-45° \sim 0°$.

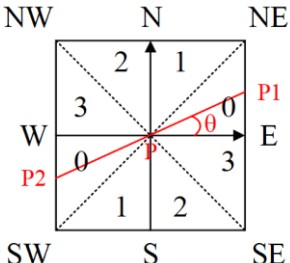

**Figure 11.** The map of the gradient and directions.

The gradient direction of pixel $p$ is theta, and the linear interpolation of the gradient of pixels $p1$ and $p2$ is:

$$\tan(\theta) = G_y / G_x \tag{17}$$

$$G_{p1} = (1 - \tan(\theta)) \times E + \tan(\theta) \times NE \tag{18}$$

$$G_{p2} = (1 - \tan(\theta)) \times W + \tan(\theta) \times SW \tag{19}$$

Therefore, the pseudo-code for non-maximum suppression is described as follows:

*if $G_p \geq G_{p1}$ and $G_p \geq G_{p2}$*
  *$G_p$ may be an edge*
*else*
  *$G_p$ should be sup pressed*

It is not important how to mark the direction. The key point is that the calculation of gradient direction should be consistent with the selection of the gradient operator.

4.  Apply Double-Threshold Detection to determine true and potential edges.

After applying Non-Maximum Suppression, the remaining pixels can more accurately represent the actual edges of the image. However, there are still some edge pixels due to

noise and color changes. In order to solve these spurious responses, it is necessary to filter edge pixels with weak gradient values and retain edge pixels with high gradient values, which can be achieved by selecting high and low thresholds.

The pseudo-code for Double-Threshold Detection is described as follows:

$$if\ G_p \geq HighThreshold$$
$$G_p\ is\ an\ strong\ edge$$
$$else\ if\ G_p \leq LowThreshold$$
$$G_p\ is\ an\ weak\ edge$$
$$else$$
$$G_p\ should\ be\ suppressed$$

5. Finally, edge detection is completed by suppressing isolated weak edges (low threshold points).

The pseudo code for suppressing isolated low threshold points is described as follows:

$$if\ G_p == LowThreshold\ and\ G_p\ connected\ to\ a\ strong\ edge\ pixel$$
$$G_p\ is\ an\ strong\ edge$$
$$else$$
$$G_p\ should\ be\ sup\ pressed$$

Through the above 5 steps, edge extraction based on the Canny algorithm can be completed.

### 2.5.2. Contour Detection

In 1985, Satoshi Suzuki introduced two algorithms to achieve contour extraction of binary images [36]. The function "findContours" in OpenCV [37] is implemented based on the idea of this paper. Figure 12 shows the effect after using this function and Figure 13 shows a further extraction of the region of interest.

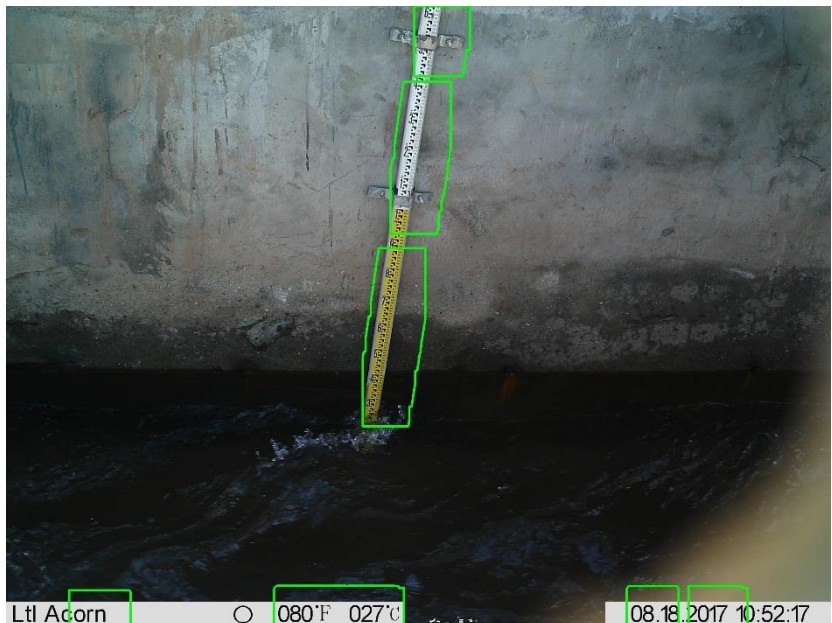

**Figure 12.** The effect of contour detection from the image in Figure 9b.

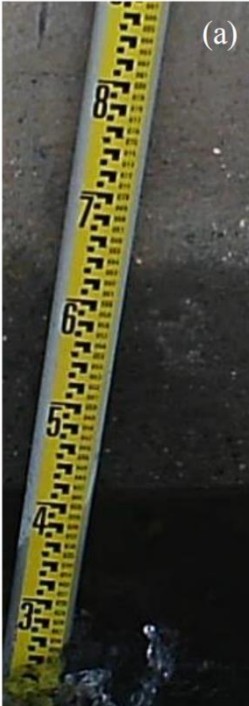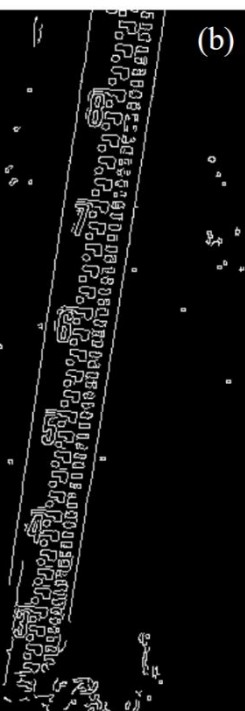

**Figure 13.** The effect of contour detection and edge detection. (**a**) is obtained from Figure 12 by the size of the area of the detected contour. (**b**) The effect from (**a**) by the edge detection of Canny operator.

### 2.5.3. Tilt Correction

Hough transform is a feature extraction technique whose purpose is to find instances of objects with specific shapes through the voting process in the parameter space [38]. The basic principle of the Hough transform is to use the duality of point and line to change the given curve in the original image space into a point in the parameter space through the curve expression. Polar coordinate parameters $(\rho, \theta)$ are used to represent a straight line, where $\rho$ is the length of the line segment, and $\theta$ is the angle between the line and the $x$ axis. To explore the $(\rho, \theta)$ parameter space, the first step is the creation of a two-dimensional histogram, and then the number of non-zero pixels of the corresponding lines and incremental arrays near the $(\rho, \theta)$ coordinates for each $\rho$ and $\theta$ value are calculated. Therefore, each non-zero pixel can be regarded as a vote for potential candidate lines. The most probable line corresponds to the parameter value that gets the most votes, that is, the local maximum in the two-dimensional histogram.

Hough line detection has the advantages of strong anti-interference ability, insensitivity to the incomplete part of the line, noise, and other co-existing non-linear structure in the image, and it is tolerant of the gap in feature boundary description and is relatively unaffected by image noise.

A slot is used to place the water ruler, which is convenient to maintain and replace damaged and contaminated water rulers. However, due to the gap between the slot and the water ruler, the water ruler is easily affected by the water flow and tilted slightly in the opposite direction. The tilt angle θ is 8.3°. As is shown in Figure 14c, in order to make the results more accurate, the length of the projection of a unit length slope on the $y$-axis is 0.87 according to the trigonometric function. The error due to tilt was calibrated in the results section by using the relationship between the angle of tilt and the value of the reading.

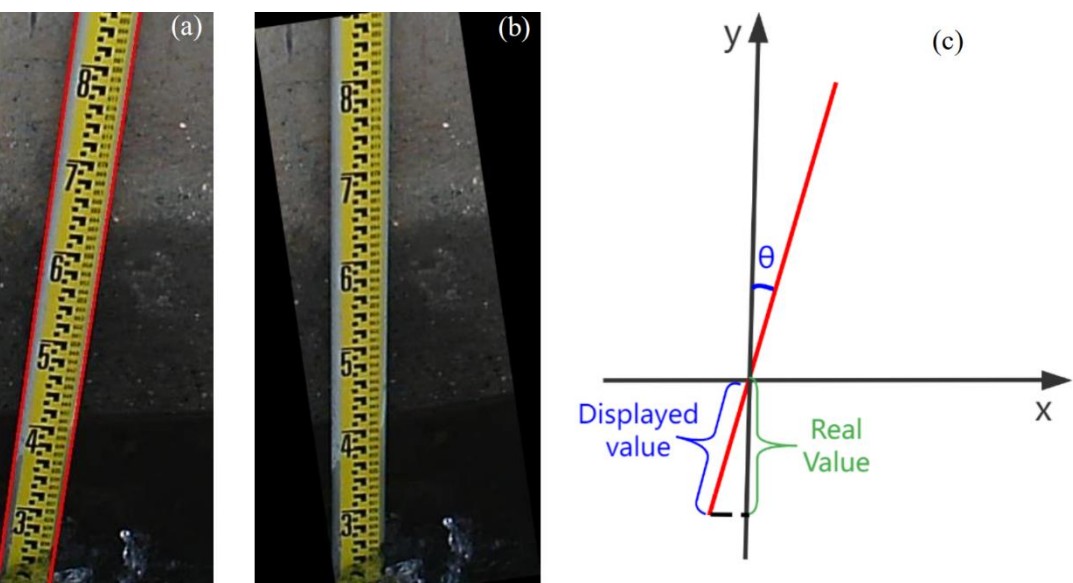

**Figure 14.** The effect of linear detection and tilt correction. (**a**) The straight line detected by Hough transform from image Figure 13b. (**b**) Rotation correction according to the detected tilt angle (8.3°) of the lines in (**a**). (**c**) The deviation between the displayed value and the real value after tilt correction.

After further cropping from Figure 14b, as shown in the processing flow in Figure 15, the part of the water ruler above the water surface is extracted by the closing operation and the function "findContours" in turn.

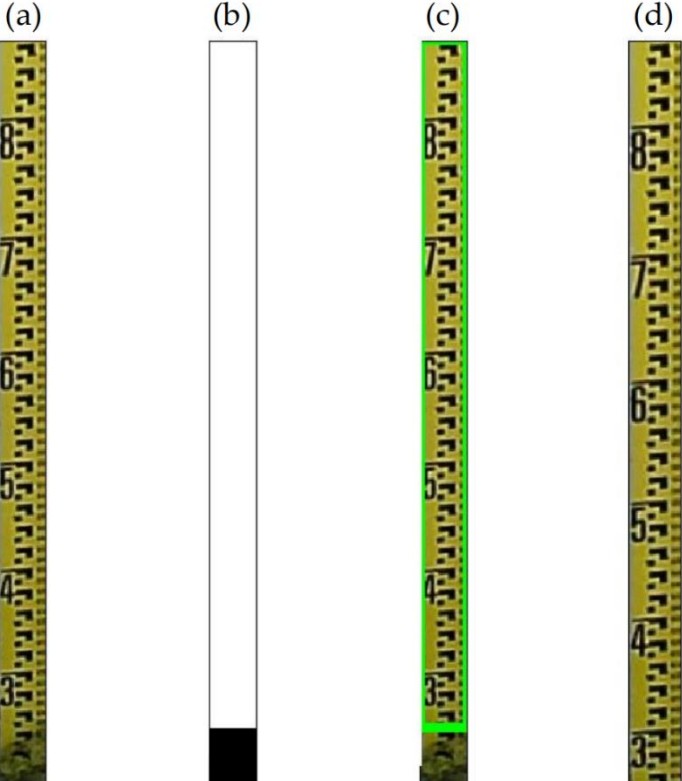

**Figure 15.** The water ruler image containing only the part above the water surface is extracted. (**a**) is the water ruler image truncated from Figure 14b. (**b**) is the effect of closing operation from (**a**). (**c**) is the effect of contour detection from (**b**) by using the method named "findContours". (**d**) is the effective area extracted from the detected contour range.

*2.6. Character Positioning and Segmentation*

The projection method [39] is used to complete the positioning and segmentation of the digital characters on the water ruler. The first stage of water level recognition is to recognize the digital characters on the water gauge, and the accuracy of cutting digital characters greatly affects the correct rate of water level recognition. The principle of the projection method is to analyze the pixel distribution histogram of the picture after binarization so as to find the dividing points of adjacent characters for segmentation. The projection of the image in the corresponding direction is to take a straight line in this direction, count the number of black dots of the pixels on the image perpendicular to the straight line (axis), and add the sum as the value of the position on the axis; The cut based on image projection is to map the image into such features and then determine the cut position (coordinates) of the image based on such features, and use this coordinate to cut the original image to get the target image.

Firstly, the binary image is horizontally projected from left to right, and the total number of black pixels in each row is calculated. The noise is then filtered and the horizontal projection is drawn to take out the numbers in each interval. The position of each numeric character is marked and returns the corresponding coordinates.

Figure 16 shows the effect of grayscale and binarization on the extracted water ruler image in Figure 15. Then as shown in Figure 17, the ruler containing the characters and the scale-lines part are split and further processed.

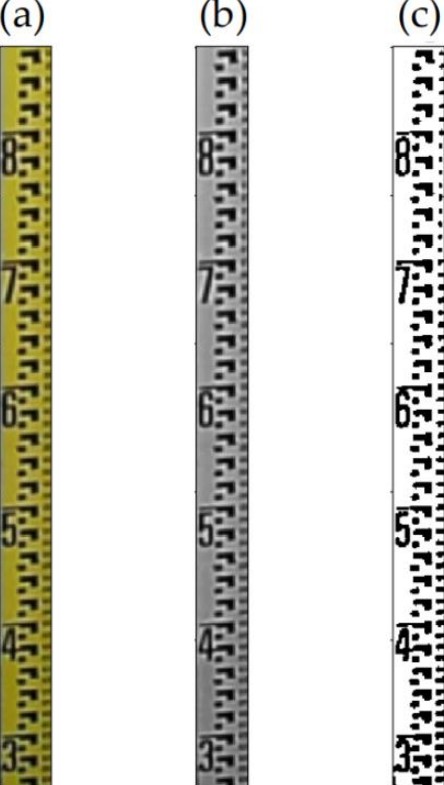

**Figure 16.** The ruler images (**a**–**c**) are, in turn, Gaussian filter, Grayscale processing, and binarization from Figure 15d.

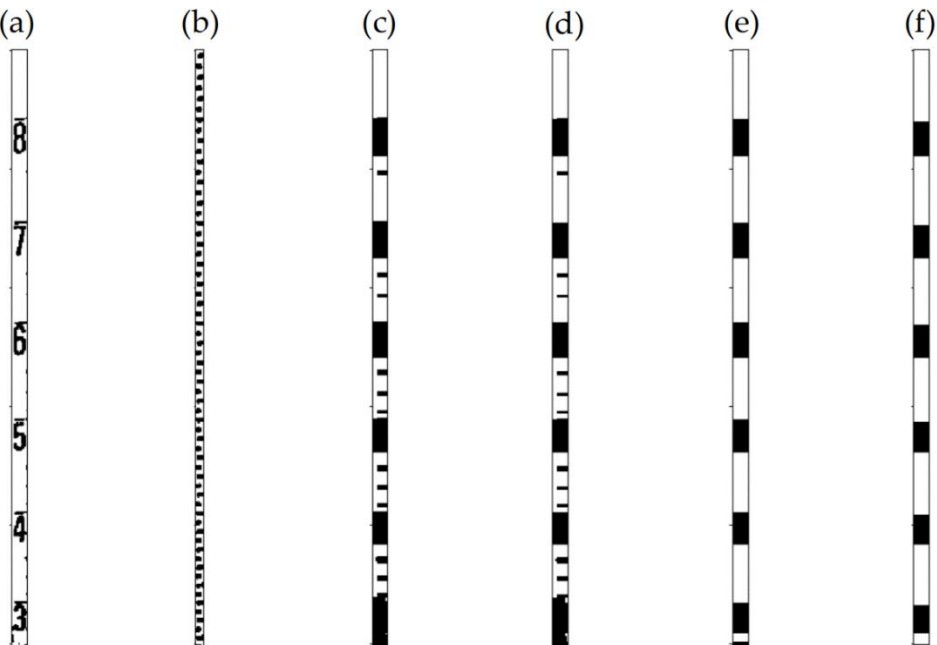

**Figure 17.** Further segmentation of characters and scale lines. (**a**) The image of the water ruler containing only one side of the array character area segmented from the image Figure 16c. (**b**) The image of the water ruler containing only one side of the scale lines area segmented from the image in Figure 16c. (**c**) The final four images (**c**–**f**) are obtained by morphological manipulation and projection from (**b**).

Figure 18 shows the segmented characters using the projection method.

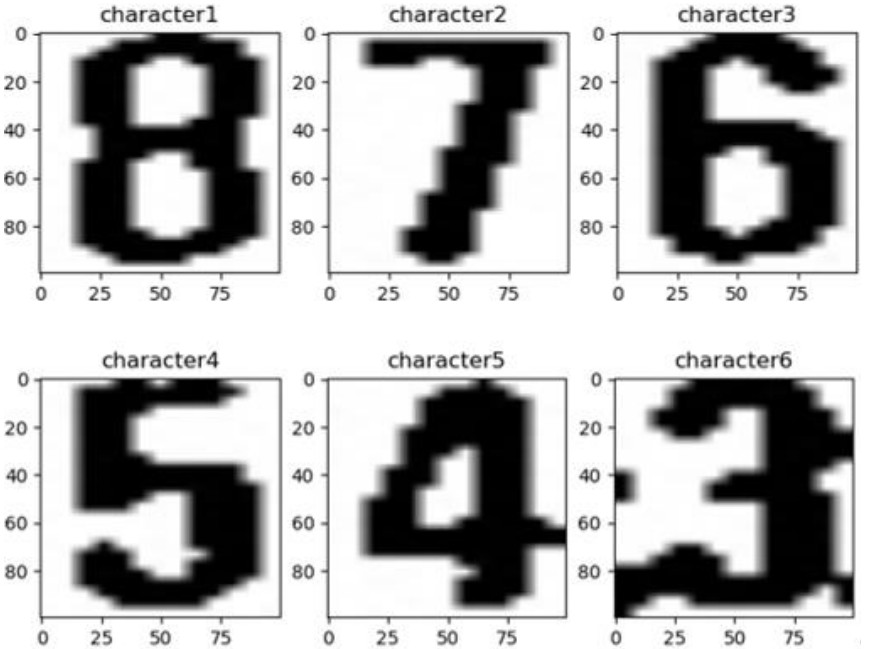

**Figure 18.** The images above are the final character images, which are positioned and divided from Figure 17f according to the projection method.

### 2.7. Identification and Calculation the Value of Water Level

This stage includes the following three steps:

1. **Recognize characters and return coordinates:** The CNN is designed to classify and recognize the segmented digital characters, take the largest character among all recognized characters, and return the position coordinates of the character.
2. **Count the number of scale lines:** A counter that counts down the scale line based on the coordinate position of the largest recognized numeric character (after a series of preprocessing operations is set up, and the pixels are traversed and counted using the pixel variation of the binary image).
3. **Calculate the value of water level:** The value of the largest numeric character identified in step (1) is used, and the value of the counter in step 2 (the value of the number of tick marks traversed) is used, which is the final water level value.

### 2.7.1. Design of CNN

To recognize the digital characters from the water gauge image, the CNN is used to recognize the digital character images Combined with the number of scale lines identified in the following text, the intelligent water level identification of the water gauge image can be completed. For the two-dimensional properties of a digital character binary image, the parameters of the CNN are determined as follows.

Design a CNN with three convolutional layers, three pooling layers, a fully-linked layer and a Dropout layer is added to prevent over-fitting.

A tensor-board is used to visualize the network structure and parameters as shown in the Table 1:

**Table 1.** The structure and parameters of the convolutional neural network designed in this study.

| Layer (Type) | Output Shape | Param |
|---|---|---|
| sequential (Sequential) | (None, 28, 28, 3) | 0 |
| rescaling_1 (Rescaling) | (None, 28, 28, 3) | 0 |
| conv2d (Conv2D) | (None, 28, 28, 16) | 448 |
| max_pooling2d (MaxPooling2D) | (None, 14, 14, 16) | 0 |
| conv2d_1 (Conv2D) | (None, 14, 14, 32) | 4640 |
| max_pooling2d_1 (MaxPooling2D) | (None, 7, 7, 32) | 0 |
| conv2d_2 (Conv2D) | (None, 7, 7, 64) | 18,496 |
| max_pooling2d_2 (MaxPooling2D) | (None, 3, 3, 64) | 0 |
| dropout (Dropout) | (None, 3, 3, 64) | 0 |
| flatten (Flatten) | (None, 576) | 0 |
| dense (Dense) | (None, 10) | 73,856 |
| dense_1 (Dense) | (None, 10) | 1290 |

**Input Layer:** The number of nodes in the input layer of a CNN is determined by the dimension of the input vector. The binarized image dimension of the digital characters to be recognized in this research is $28 \times 28$, so the number of nodes in the input layer is 784.

$$Param = (k_w \times k_h \times Ch_s + 1) \times k_n \tag{20}$$

$k_w$ is the width of convolution kernel;
$k_h$ is the height of convolution kernel;
$k_n$ is the number of convolution kernels;
$Ch_s$ is the number of channels for input data.
**Convolution Layer**:
The first convolutional layer: Conv2D (16, 3, padding = 'same', activation = 'relu').

$$Param = (3 \times 3 \times 3 + 1) \times 16 = 448 \tag{21}$$

The second convolutional layer: Conv2D (32, 3, padding = 'same', activation = 'relu'). After 16 convolution kernels in the first convolution layer, the number of channels of input data in the second convolution layer becomes 16.

$$aram = (3 \times 3 \times 16 + 1) \times 32 = 4640 \tag{22}$$

The third convolutional layer: Conv2D (64, 3, padding = 'same', activation = 'relu'). After 32 convolution kernels in the second convolution layer, the number of channels of input data in the second convolution layer becomes 32.

$$Param = (3 \times 3 \times 32 + 1) \times 64 = 18496 \tag{23}$$

**Pooling Layer:**

The fourth layer of the network is the first maximum pooling layer. The default parameters for this layer is the pooling size of 2 and the fill method of valid padding. Pooling is used to reduce the dimension of the data, so the parameter is 0. After the data goes through the pooling layer, the dimension is reduced to half of the original, and the data output dimension is (14, 14, 16).

The sixth layer of the network is the first maximum pooling layer. The default parameter of this layer is pooling size 2, valid padding and the data output dimension is (7, 7, 32).

The eighth layer of the network is the first maximum pooling layer. The default parameter of this layer is pooling size 2, valid padding and the data output dimension is (3, 3, 64).

**Dropout Layer:**

When using a deep learning model, the two problems of over-fitting and under-fitting must be considered [40]. Among them, the methods to solve under-fitting include increasing the data set, optimizing the model, etc., which are treated according to specific problems. The problem of over-fitting can be solved by Dropout shown in Figure 19 to improve the generalization ability of the model. Using Dropout in a neural network is actually abandoning a part of the connection during forwarding propagation, so that some neurons do not work, which can improve the generalization ability and prevent excessive dependence on local features [41].

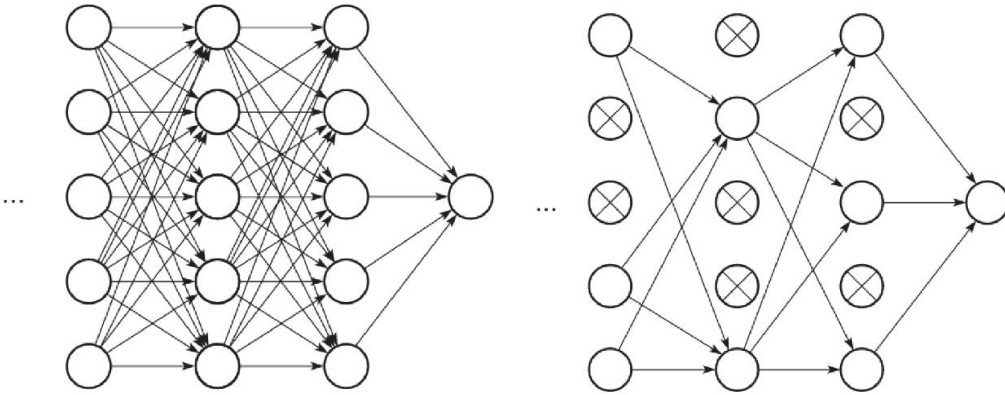

**Figure 19.** The Fully Connected Layer part of a neural network (**left**) and the Fully Connected Layer part after adding the Dropout layer (**right**).

**Flatten Layer:** It is used to "Flatten" the input data, that is, to make the multidimensional input one-dimensional. It is commonly used in the transition from the convolutional layer to the fully connected layer, and the Flatten layer does not affect the size of the batch.

The Flatten layer is placed between the convolution layer and the full connection layer to play a role in transformation. Because the output result of the convolution layer is a two-dimensional tensor, multiple feature graphs will be outputted after passing through

the convolution layer. Only when these feature graphs are converted into the form of vector sequence can they be one-to-one corresponding to the full connection layer.

**Fully Connected Layer:** There are two layers, the Param of the fully connected layer neural network, which describes the number of neuron weights in each layer. Its calculation formula is as follows:

$$Param = (Ch_s + 1) \times k_n \tag{24}$$

$k_n$. is the number of convolution kernels;

$Ch_s$ is the number of channels for input data.

The first fully connected layer is Dense (128, activation = 'relu'), and through the action of Flatten, the number of channels of output data becomes 576, and there are 128 convolution kernels in Dense, so $Param = (576 + 1) \times 128 = 73856$.

The second fully connected layer is Dense (10), and the number of channels of output data is 128, and there are 10 convolution kernels in Dense, so $Param = (128 + 1) \times 10 = 1290$.

**Output Layer:**

Since this article recognizes a total of 10 Arabic numerals from 0–9, the number of nodes in the output layer is 10.

2.7.2. Train the CNN

1.  Ten image samples are selected containing printed numeric characters 0–9, each containing 1016 binary images with a size of $128 \times 128$, for a total of 10,160 numeric character images. We randomly assign 80% of the training set and 20% of the validation set to be the data set to train the CNN.

2.  After 50 epochs of iterative training, the training results show that when the loss function converges, the recognition accuracy of the neural network on the verification set reaches 97-98%, which is shown in Figure 20.

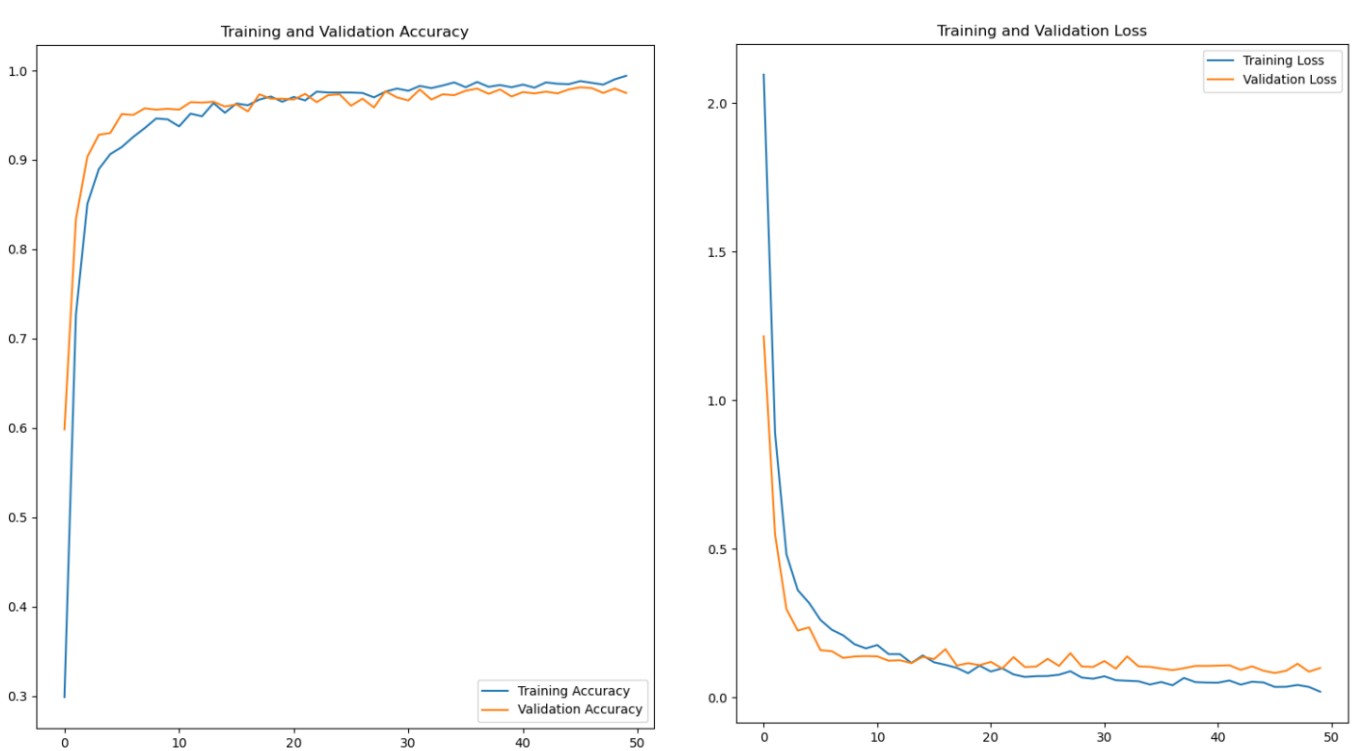

**Figure 20.** The accuracy (**left**) and loss (**right**) on training and validation.

3.  Save the best training results as h5 model, evaluate the model and call it in the test phase.

*2.8. Extraction of Scale Line and Calculation of Water Level*

In this section, a counter is set up to mark the number of scale lines, starting with N = 0, starting with the digital character coordinates with the maximum value *M* identified in the above experimental step. The detailed flow is shown in Figure 21.

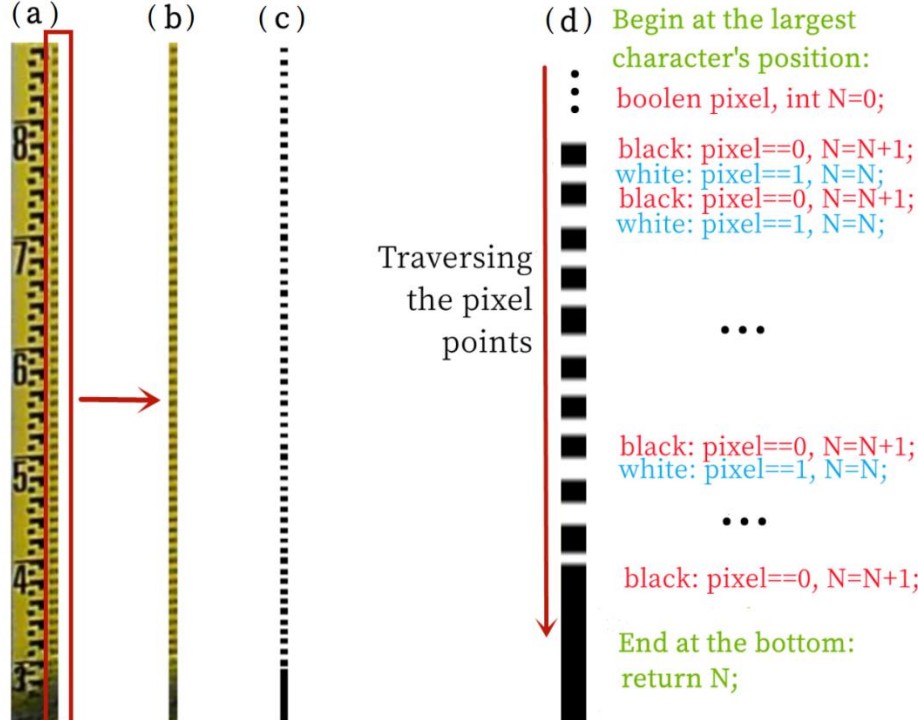

**Figure 21.** Flowchart and pseudo-code for calculating the number of scale lines. (**b**) is the portion of the scale extracted from (**a**). (**c**) is the binarized image of (**b**). (**d**) is the pseudo-code of the calculation procedure to obtain the number of scale-lines.

Going down through the pixels of the water ruler image after the binarization, the value *N* of the counter increases by 1 for every change of the pixel value, and the final value of the counter is the number of scale lines detected.

Thus, the calculation formula of the final water level value is:

$$WL = (M \times 10 - N)/100 \ (m) \tag{25}$$

**3. Results**

In order to meet the requirement of using the manually observed water level value as the standard value for the algorithm accuracy assessment, the images with good lighting conditions were selected from the photographs taken within fifteen days from 3 August 2017 to 18 August 2017, and 40 of them were randomly selected for the control experiment. In order to evaluate the performance of this method in terms of accuracy and speed, the experimental results of the template matching algorithm are added as a comparison. The template matching algorithm was one of the most representative methods in image recognition prior to the rise of CNN. It extracts a number of feature vectors from the image to be recognized and compares them with the corresponding feature vectors in the template library, calculates the distance between the image and the template feature vectors, and determines the category to which the image belongs by the minimum distance method. The manual reading, template matching algorithm, and intelligent recognition algorithm proposed in this study are used for control experiments.

The research hardware configuration is a server equipped with an Intel i7-10875H, 2.3 GHz processor, 16G running memory and an NVIDIA GeForce RTX 2060 graphics

card. After the same image pre-processing process, the computational accuracy results of the template matching algorithm and the intelligent recognition algorithm designed in this study were compared under the same data set as shown in Table 2: Forty sets of data were selected for comparison, including the results of the manual readings, the template matching algorithm and the intelligent recognition algorithm proposed in this study. Using the manual readings as the standard value for reference, the average error rate of the template matching algorithm reaches 28.98%, while the average error rate of the intelligent recognition algorithm is only 5.43%. It can be seen that the accuracy rate of the intelligent recognition algorithm proposed in this study is approximately 94–95%, which is nearly 24% higher than that of the template matching algorithm. As shown in Figure 22., the calculation result of the intelligent recognition algorithm almost matches the line of manual visual reading with an average error of only 0.01m, the calculation error is relatively stable and the calculation result is highly close to the real value; while the average error of the template matching algorithm is 0.07m, which shows a large error and instability compared with the real value. In terms of time loss as shown in Figure 23, under the same hardware configuration mentioned above, each group of tests proved that the time loss of the intelligent recognition algorithm was less than that of the template matching algorithm. Among them, the average time consumption of the template matching algorithm is 8.907s, while the average time consumption of the intelligent recognition algorithm is only 4.741s. The intelligent recognition algorithm designed in this study saves 46.77% in time loss compared with the template matching algorithm.

**Table 2.** The results of the comparison between the template matching algorithm and the intelligent recognition algorithm designed for this experiment. The accuracy and time loss of these two algorithms were compared using manual readings as the standard. Each of the manual readings is the average of three manual readings.

| Ruler No | Visual (m) | Template Matching | | | Intelligent Recognition | | |
|---|---|---|---|---|---|---|---|
| | | Value (m) | Error | Time (s) | Value (m) | Error | Time (s) |
| 1 | 0.23 | 0.22 | 4.35% | 8.32 | 0.23 | 0.00% | 6.94 |
| 2 | 0.27 | 0.10 | 62.96% | 8.44 | 0.28 | 3.70% | 6.82 |
| 3 | 0.24 | 0.24 | 0.00% | 8.73 | 0.24 | 0.00% | 4.59 |
| 4 | 0.24 | 0.06 | 75.00% | 8.96 | 0.23 | 4.17% | 4.58 |
| 5 | 0.24 | 0.16 | 33.33% | 9.29 | 0.24 | 0.00% | 4.58 |
| 6 | 0.26 | 0.17 | 34.62% | 9.74 | 0.25 | 3.85% | 4.58 |
| 7 | 0.23 | 0.17 | 26.09% | 9.35 | 0.17 | 26.09% | 4.56 |
| 8 | 0.24 | 0.10 | 58.33% | 8.93 | 0.19 | 20.83% | 4.6 |
| 9 | 0.23 | 0.19 | 17.34% | 8.78 | 0.19 | 17.39% | 4.61 |
| 10 | 0.22 | 0.20 | 9.01% | 9.48 | 0.21 | 4.55% | 4.61 |
| 11 | 0.23 | 0.17 | 26.09% | 8.89 | 0.25 | 8.70% | 4.6 |
| 12 | 0.24 | 0.07 | 70.83% | 8.26 | 0.24 | 0.00% | 6.86 |
| 13 | 0.24 | 0.14 | 41.67% | 9.27 | 0.23 | 4.17% | 4.53 |
| 14 | 0.24 | 0.16 | 33.33% | 8.92 | 0.24 | 0.00% | 4.63 |
| 15 | 0.23 | 0.22 | 4.35% | 9.32 | 0.23 | 0.00% | 4.66 |
| 16 | 0.22 | 0.19 | 13.64% | 9.63 | 0.19 | 13.64% | 4.59 |
| 17 | 0.21 | 0.13 | 38.10% | 8.64 | 0.22 | 4.76% | 4.44 |
| 18 | 0.23 | 0.14 | 39.13% | 9.33 | 0.23 | 0.00% | 4.58 |
| 19 | 0.27 | 0.22 | 18.52% | 8.28 | 0.27 | 0.00% | 4.37 |
| 20 | 0.28 | 0.30 | 7.14% | 8.72 | 0.30 | 7.14% | 4.61 |
| 21 | 0.26 | 0.16 | 38.46% | 8.37 | 0.24 | 7.69% | 4.56 |
| 22 | 0.28 | 0.17 | 39.29% | 8.88 | 0.26 | 7.14% | 4.58 |
| 23 | 0.27 | 0.26 | 3.70% | 8.96 | 0.26 | 3.70% | 4.6 |
| 24 | 0.21 | 0.22 | 4.76% | 8.92 | 0.21 | 0.00% | 4.61 |
| 25 | 0.23 | 0.25 | 8.70% | 8.94 | 0.23 | 0.00% | 4.63 |
| 26 | 0.21 | 0.19 | 9.52% | 8.95 | 0.19 | 9.52% | 4.6 |
| 27 | 0.21 | 0.11 | 47.62% | 9.04 | 0.20 | 4.76% | 4.6 |

**Table 2.** *Cont*.

| Ruler No | Visual (m) | Template Matching | | | Intelligent Recognition | | |
|---|---|---|---|---|---|---|---|
| | | Value (m) | Error | Time (s) | Value (m) | Error | Time (s) |
| 28 | 0.20 | 0.10 | 50.00% | 9.1 | 0.18 | 10.00% | 4.58 |
| 29 | 0.22 | 0.12 | 45.45% | 9.12 | 0.23 | 4.55% | 4.58 |
| 30 | 0.21 | 0.11 | 47.62% | 8.09 | 0.20 | 4.76% | 4.68 |
| 31 | 0.23 | 0.30 | 30.43% | 8.8 | 0.24 | 4.35% | 4.64 |
| 32 | 0.20 | 0.11 | 45.00% | 8.21 | 0.20 | 0.00% | 4.36 |
| 33 | 0.23 | 0.22 | 4.35% | 9.01 | 0.21 | 8.70% | 4.63 |
| 34 | 0.22 | 0.21 | 4.55% | 8.17 | 0.22 | 0.00% | 4.47 |
| 35 | 0.21 | 0.21 | 0.00% | 8.38 | 0.21 | 0.00% | 4.19 |
| 36 | 0.20 | 0.18 | 10.00% | 9.09 | 0.19 | 5.00% | 4.63 |
| 37 | 0.21 | 0.14 | 33.33% | 9.21 | 0.23 | 9.52% | 4.62 |
| 38 | 0.21 | 0.10 | 52.38% | 9 | 0.19 | 9.52% | 4.57 |
| 39 | 0.21 | 0.12 | 42.86% | 9.81 | 0.21 | 0.00% | 4.59 |
| 40 | 0.22 | 0.16 | 27.27% | 8.94 | 0.24 | 9.09% | 4.59 |
| Average | — | — | 28.98% | 8.91 | — | 5.43% | 4.74 |

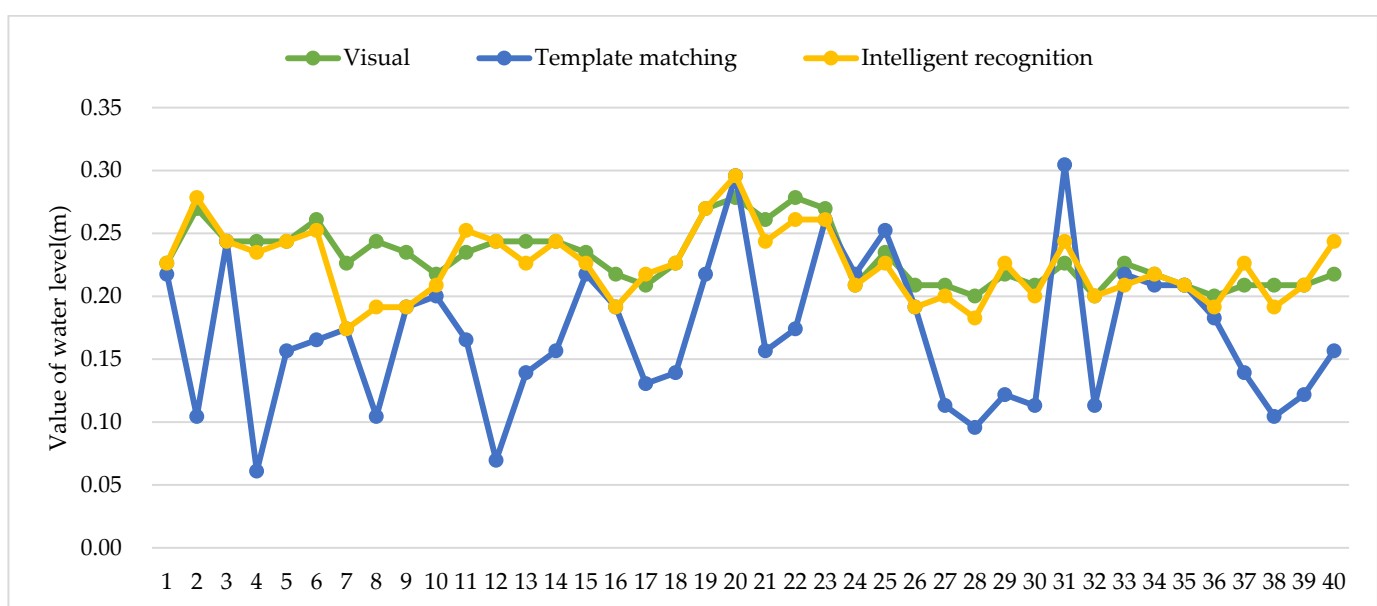

**Figure 22.** Comparison of water level values. The green line shows the values by visual reading. The yellow line represents the values of intelligent recognition. The bule line represents the values by the method of template matching. The abscissa represents the sequence number of the image and the ordinate represents the value of the reading (the unit is m).

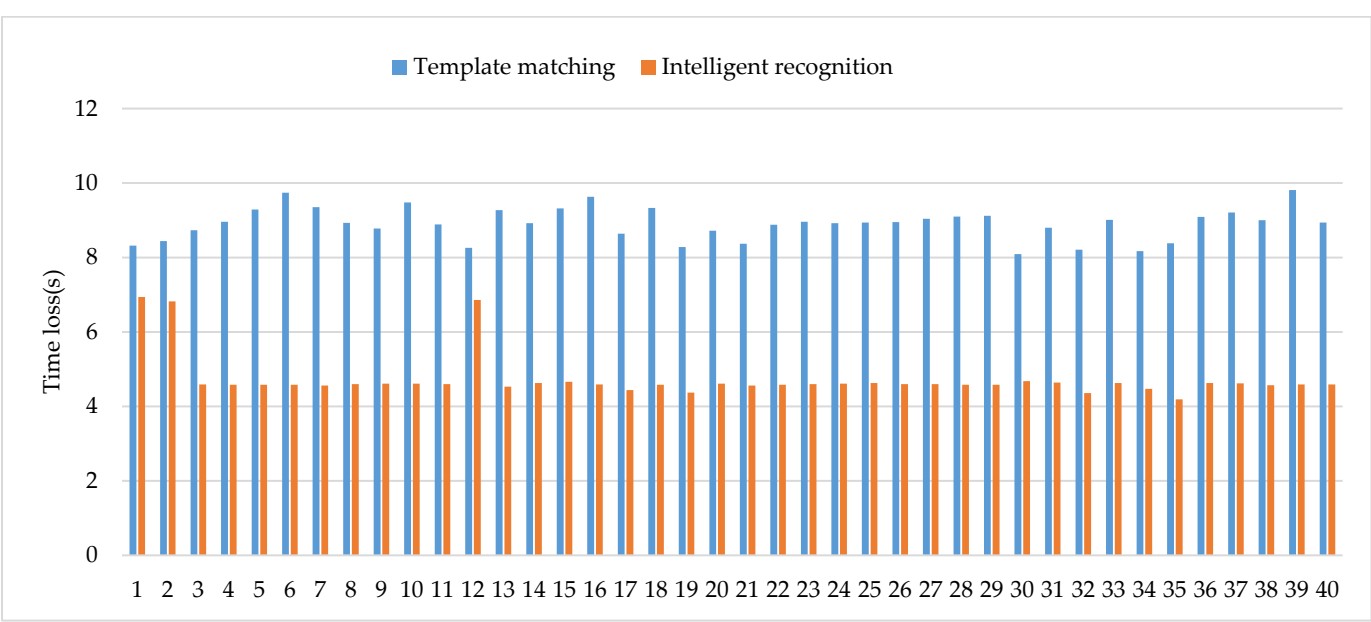

**Figure 23.** Comparison of time loss. The orange columns show the time loss of intelligent recognition. The blue columns represent the template matching. The ordinate represents the sequence number of the image and the abscissa represents the value of the time loss (the unit is s).

## 4. Discussion

This paper proposed a new reading recognition process of the water gauge image. First, position calibration was carried out on a photo with good illumination and a low water level, which can cover all scales and digital characters to the maximum extent, which is conducive to data set expansion and image batch segmentation. Next, in the stage of scale lines extraction, the tick mark counting method based on pixel traversal of the binary image has more stable performance and higher accuracy. Finally, the CNN is designed in the intelligent recognition stage, and the template matching algorithm is reproduced in this stage so that the two can be compared based on the same training set and test set.

Various sensor-based water level identification methods have obvious disadvantages, such as high cost, difficulty in deployment, easily damaged and difficult to promote [42–47]. Takagi et al. proposed a water level detection algorithm based on video signals, but it is difficult to install a measuring board such as a wooden board in many waters [48]. Sun et al. used Wiener filtering, morphological transformation, edge extraction and template matching algorithm based recognition methods to process the water level recognition of boiler water meters, and finally calculated the value of the water level [49]. Although the method was low-cost and efficient at the time, this study introduces a more intelligent processing flow and recognition algorithm. The results show that the recognition accuracy of the CNN based on the deep learning algorithm is improved by 23.55% compared to that of the template matching algorithm on the same data set. This method has lower cost and maintainability compared to the automatic water level meter. While ensuring a higher accuracy, its error is also within the acceptable range for water level monitoring. In addition, in the field of water science research, if the area and flow rate of a hydrological section is known, then the flow rate value can be calculated based on the water level value intelligently identified by this system. Therefore, this study is important in that it can help to realize the intelligent monitoring of flow.

However, this research also needs to be improved. For example, although the digital character recognition stage has achieved high accuracy, the recognition rate is low in the case of dark light, resulting in the difficulty of determining the threshold in the preprocessing stage. Therefore, it needs a significant amount of manual debugging to find the appropriate threshold so as to better carry out image segmentation. In addition, although the whole

process is reasonable, the procedure needs to be adjusted when different kinds of rulers are encountered, so further improvement is needed in terms of applicability. Furthermore, since the study area selected for this method is a mountainous watershed, there is still a lack of application studies in urban catchments, but since urban environments are more stable than mountainous areas with more standardized study areas, this method has the potential to be extended to urban catchments.

## 5. Conclusions

In this study, we proposed a method that used image processing technologies and CNN to identify the water level values in water scale images. The source image of the water ruler taken automatically by the camera at the hydrographic section is used as the input, and is pre-processed by image processing techniques. The characters and the location coordinates of the water ruler are calibrated and extracted in batches, and then the CNN is used for intelligent recognition of the digital characters. The number of scale lines located below the biggest identified character is obtained by traversing the pixel points of the binarized image downward from the coordinates of this numeric character according to the pattern of pixel value changes. Finally, the value of the water level is calculated using the mathematical relationship between the value of the character and the number of scale lines on the water ruler.

This method is more intelligent, low cost, and easy to maintain for the whole process from image acquisition to recognition to the calculation of the water level value. The study shows that the accuracy of this method can meet the requirements of hydrological monitoring. Using this method, water level monitoring in mountainous watersheds can be achieved.

**Author Contributions:** Formal analysis, R.C.; Methodology, G.D.; Writing, G.D. and Z.L.; Data curation, G.D. and C.H.; Investigation, Z.L. and J.L. Software, G.D.; Data analysis, G.D. and Z.L.; Project administration, R.C.; Funding acquisition, R.C., C.H., Z.L. and J.L. All authors have read and agreed to the published version of the manuscript.

**Funding:** This research was funded by the National Key Research and Development Project of China (2019YFC1510505) and the National Natural Science Foundation of China (42171145, 41971041, 42171147, 41877163).

**Institutional Review Board Statement:** Not applicable.

**Informed Consent Statement:** Not applicable.

**Data Availability Statement:** Numerical results reported in this paper maybe shared by the interested parties if requested. Please contact the author.

**Acknowledgments:** The authors thank all the colleagues participating in the Hei project for their sharing data and support.

**Conflicts of Interest:** The authors declare that they have no conflict of interest.

## Abbreviations

| | |
|---|---|
| CNN | Convolutional Neural Networks |
| ANN | Artificial Neural Networks |
| CV | Computer Vision |
| SVM | Support Vector Machine |
| ILSVRC | ImageNet Large Scale Visual Recognition Challenge |
| CNY | China Yuan |
| DTU | Data Transfer Unit |

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
