# Peer review of "Research on Water-Level Recognition Method Based on Image Processing and Convolutional Neural Networks"

_water, doi:10.3390/w14121890_

Round 1
Reviewer 1 Report
Comments on paper entitled" Research on Water-Level Recognition Method Based on Image Processing and Convolutional Neural Networks" submitted by Dou et al. to the special issue "Application of AI and UAV Technics in Urban Water Science" of Water.
General Comments
This paper explores the application of computer science on the field of water science, and the authors propose an intelligent method for the recognition of water ruler image readings based on image processing techniques and CNN. Firstly, digital image processing algorithms such as grayscale transform, edge detection, tilt correction based on Hough transform and projection method are used to extract and pre-process the interest region of the source image. Then the CNN is designed and trained to recognize the segmented digital characters. Finally, the value of water level is calculated based on the quantitative relationship between the value of the recognized digital characters and the number of detected tick marks. In general, this is a good work for the field of hydrology, especially for the intelligent management of water resources and it is necessary to be published in Water soon. Only following specific comments need the authors to be paid attention to.
Specific comments
1. The section “Abstract” of the article requires about 200 words maximum, so please streamline the original text again.
Please check the formatting of citations in the text, especially when citing an article with multiple authors, for example, Line 67.
- Please revise the format of the figures in the article. If there are multiple panels, they should be listed as: (a) Description of what is contained in the first panel; (b) Description of what is contained in the second panel. A caption on a single line should be centered.
- Please check the description of the title of subsection 2.5.1. in Line 239, the word “Graying” is used incorrectly.
- Please streamline the content of subsection 2.5.1. Only the logic and principles of the edge detection algorithm need to be clearly stated, and some specific examples of them do not need to be written. Lines 239-345.
- The language of the part entitled “Tilt correction” needs to be well organized again. Lines 366-390.
- A space is required between the number and the unit in the description of the frequency of the processor. Line 559.
Author Response
Please see the attachment.
This document is a response to your comments. Due to the limitation that the system can only upload one file, some changes cannot be reflected in this document, so more details are in the manuscript and we look forward to your response.
Thank you again for your careful reading and valuable comments.

Reviewer 2 Report
Please find the attached file.

Author Response

(The authors gave the same response as above.)

Reviewer 3 Report
The issue of automating the process of water discharge determination in watercourses is very important. For this purpose, the authors used a vision system recording the water level on a constructed water gauge. Certainly then the presented topic is within the scope of the journal.
After reading the manuscript many doubts arise, from the assumptions made, collected data, to data analysis. Instead of describing research in detail, the authors focus on presenting the elementary concepts of computer vision and machine learning.
It is impossible to perform a similar analysis based on photos collected just during three days. In my opinion, it is necessary to collect a much larger number of photos for different discharges and weather conditions.
I did not found an answer to the key question - how did the authors estimate the water level? "Count the number of scale lines" does not convince me, because we still do not know what accuracy can we expect for different water table levels and under different lighting.
For these reasons, I believe that the manuscript needs thorough rethinking and redrafting.
Major comments
- Key results
I am not able to answer the question of what is the key result of the work, because the presented analysis does not have solid foundations. - Originality and significance
The authors described the secondary elements in great detail, leaving a lot of understatement in relation to the key issues. The most original feature of this work is probably the custom convolutional neural network designed for text recognition. However, it is also the weakest element of the whole work, as modern machine learning models are significantly better in this area. - Clarity and context
The abstract is very vague and does not emphasize the novel elements of the manuscript. Describing and using many methods of image analysis without referring to other authors and justifying their use rather makes it difficult to understand the proposed methodology. Many parts of the text read a bit like the basics of computer vision and machine learning. The term "segmentation" should be clearly defined. It means something different in machine learning and image analysis. - Data and methodology
What discharge rates are expected at constructed gauge station? Have the authors tried to estimate this? I assume that such data was the basis for the design of this waterbed. It is not described how Data Transfer Unit works (Figure 3).
There is no information in the text about the camera and photos. It is not known how many photos were used for the analysis. Most of the information on the data used is at the very end of the document, instead of at the beginning (Materials and Methods section).
It is not known what IT tools the authors used. One can only guess that this is the OpenCV library (mentioned in one sentence) for computer vision section. For such projects, it is standard to make the program code and models available in public repositories.
Incorrect terms are used in several places. "After 50 rounds of iterative training ..." Those rounds are called epochs.
Figures are of very low quality. It is about both their graphic quality, content and the lack of description of the axis.
Workflow presented in Figure 4 raises many questions and is unclear. It would be difficult to describe how exactly the authors obtain the final result.
I do not understand the key concept of scale lines counting.
<lines 434-435> "Count the number of scale lines: Set up a counter that counts down the scale line based on the coordinate position of the largest recognized numeric character ..." Why do we refer to the largest recognized character if the ruler has numbers from bottom up (Figure
15)?
I may be wrong, but from my experience and from what can be seen in Figure 11, this ruler will be destroyed during the first flood. - Statistics and uncertainties
With such a small number of observations and the practically constant height of the watertable, the calculated statistics will be unreliable.
It is also impossible to estimate the sensitivity of the proposed model without different discharge, lighting, weather conditions (rain, fog), water transparency. - Conclusions
There was no clear description of the reference method in the text - Template matching. For this reason, the presented conclusions are difficult to interpret.
The conclusions regarding the manual observations are correct, but in my opinion they are based on insufficient data. For this reason, they cannot be considered credible. - References
Some references to literature are very confusing. Quoted in the first sentence [1] does not refer to the issues suggested by the authors.
Some references are incomplete [25,26,27,30] or inaccessible (in Chinese?) [22, 31].
There is no reference to similar works by other authors. Here are some examples:
Sabbatini, L., Palma, L., Belli, A., Sini, F., & Pierleoni, P. (2021). A Computer Vision System for Staff Gauge in River Flood Monitoring. Inventions 2021, 6, 79.
Jafari, N. H., Li, X., Chen, Q., Le, C. Y., Betzer, L. P., & Liang, Y. (2021). Real-time water level monitoring using live cameras and computer vision techniques. Computers & Geosciences, 147, 104642.
Detailed comments
- <line 10> Water level dynamics in catchment-scale rivers ...
How the water level can be analyzed at catchment scale? - <lines 15-18> In this method, to carry out batch segmentation of the source image, the position of the ruler and the digital characters on the ruler in the source image are demarcated by using edge detection, gray processing, Hough-transform, tilt correction, image segmentation,
and morphological processing.
So image segmentation is used to carry out batch segmentation? This one sentence is too complicated. - <21-22> This method is used for recognition after image acquisition ...
Recognition of what? - <58-60> The classiffication accuracy of these algorithms is getting higher and higher, and the training speed is getting faster and faster, which makes the field of image processing and computer vision boom.
This sentence must be rewritten. - <60-62> Artificial intelligence algorithms like decision trees, support vector machines, K-nearest neighbor algorithms, neural networks, etc., which are hot new methods in recent years...
The presented list of terms cannot be compared in this way. - <83> ... the elevation fluctuates from 2960 to 4820 m asl.
This sentence must be rewritten. - Figure 1. The geographical location of Hulu watershed.
This figure should have some spatial references. Informative quality of this figure is very low.
Where the described cross-section is located? At the watershed outlet in the north? This should be clearly presented. - Figure 2.
The dimensions of this hydrotechnical construction are given in a selective way, and they contain one obvious error. It is a pity that the location of the camera and the gauge ruler were not marked. - <112-114> In this study, for the same group of photos, because the camera is fixed, the picture frame is also fixed, therefore, the size of the same group of photos and the position of the water ruler in the photo is also fixed.
I don't understand this sentence.
From this point on, I decided not to write detailed comments as there are too many of them.
Author Response

(The authors gave the same response as above.)

Round 2
Reviewer 2 Report
I believe it is good for publication now.
Author Response
Thank you for reading and comments.
Please see the attachment.

Reviewer 3 Report
I thank the authors for referring to my comments in the revised version of the article. Several points are now more understandable to the reader.
The individual paragraphs describe the fragments of the applied workflow.
Each one of them is understandable.
The problem appears when we try to answer the question from a broader perspective - what specific problem did the authors want to solve?
Unfortunately, some of the added text fragments, instead of narrowing down the problem, blur it even more than it was in the original version.
Taking into account the three aspects included in the presented method, only the one related to computer vision seems complete and justified.
The hydrological aspect is difficult to defend, because apart from general statements, there are no hydrological data or analyzes, even the basic ones.
The machine learning aspect relates to the area of character recognition. The results were not confronted with state-of-the-art character recognition models. Why did the authors not use one of them directly as pre-trained model?
After reading the entire text, the following problems arise:
Problem 1.
"Using automatic water level sensors has disadvantages such as high cost, low accuracy and difficult maintenance. In this study, a water level recognition method based on digital image processing technology and CNN is proposed."
In which of these three points is the proposed method better?
Is the proposed method cheaper? I do not think so, but I would love to see the estimated costs.
Does it offer better accuracy? This would need to be demonstrated.
The results were compared only to the template matching algorithm, not related to automatic water level sensors.
Is the computer vision infrastructure easier to maintain comparing to automatic sensors? This should be justified on the basis of long-term observations.
Problem 2.
A problem that I didn't notice before. The water ruler is tilted in the plane of the cross-section (Figure 2) at an unknown angle!
Tilt correction described in section 2.5.3. refers (probably) to a plane parallel to the water flow.
Is the water ruler tilted in two planes?!
What is the reason for this and what are the consequences?
Problem 3.
What is the rationale behind using machine learning to recognize text?
Isn't it easier to measure the position of individual markers once, since the camera and ruler are fixed permanently?
Paradoxically, removing the element of machine learning from the workflow and focusing on image analysis will make the proposed method more reliable. The presented method, in my opinion, uses machine learning for a completely secondary task that can be carried out better with simple methods.
State-of-the-art machine learning models are able to report directly the level of water table. The method proposed by the authors uses advanced tools for a relatively easy task for which they could use the pre-trained models.
Problem 4.
What problem does the proposed method solve?
Mountain catchments and urban catchments are extremely different environments.
There are papers in which the authors propose the use of already existing city video monitoring systems for such tasks.
But here the authors conduct research in the mountain basin, and the conclusions are extrapolated to the urban catchments.
I also asked in the first review about the estimated discharges for a reason, because you cannot write about flood protection if the analyzed only cases will result in low water levels.
The article does not provide information on the applicability ranges of the proposed method. For which flows these analyzes were made?
You can read about the advantages of the proposed method itself, but there is no information about the disadvantages or applicability of the proposed method.
Problem 5.
What is the precision of manual reading of the inclined water ruler? Looking at the Figure 12, I would say about 3-5 cm. The fluctuations of the analyzed series of 40 observations are not much larger (Figure 21). This means that the range of data variability almost coincides with the reading error.
The reader should find answers to such questions directly in the text. Too many elements need to be guessed, especially from the hydrological perspective.
Author Response

(The authors gave the same response as above.)
